# Important features of bench press performance in non-disabled and Para athletes: A scoping review

Rob Buhmann[1]*, Mark Sayers[1], Julia O'Brien[2], David Borg[3,4]

**1** School of Health and Behavioural Sciences, University of Sunshine Coast, Sunshine Coast, Australia, **2** Independent Researcher, Sunshine Coast, Australia, **3** Queensland University of Technology, School of Exercise and Nutrition Sciences, Brisbane, Australia, **4** Queensland University of Technology, Centre for Data Science, Brisbane, Australia

* rbuhmann@usc.edu.au

## Abstract

### Purpose

Understanding important features for performance in non-disabled bench press and Paralympic powerlifting may inform talent identification and transfer models. The aim of this scoping review was to describe features associated with bench press performance.

### Methods

We conducted a systematic search of three electronic databases (PubMed, SportDiscus and EMBASE) to identify studies involving non-disabled and Para athlete populations that investigated features related to bench press one-repetition maximum (1RM), across six domains (anthropometric, body composition, demographic, technical, disability and neuro-muscular). Search terms included "resistance training", "para powerlifting' and "one repetition max". No date restrictions were include in searches. Studies using adult participants, with at least six-months of bench press experience, who were able to bench press their body mass were included.

### Results

Thirty-two studies met our inclusion criteria. Twenty-four studies involved non-disabled athletes (total n = 2,686; 21.9% female) and eight involved Para athletes (total n = 2,364; 39.4% female). Anthropometric (17 studies) and body composition (12 studies) features were most studied; half of the 32 studies investigated features from a single domain. Of anthropometric variables, upper arm circumference shared the strongest association with bench press 1RM in non-disabled (r = 0.87) and para-athletes (r = 0.65). Upper limb fat free mass (r = 0.91) and body mass index (r = 0.46) were the body composition variables sharing the strongest association with bench press 1RM in non-disabled and para-athletes. Few studies considering the uncertainty of their results. Practices of open and transparent science, such as pre-registration and data sharing, were absent.

**Data Availability Statement:** All data files and code are available from https://zenodo.org/records/10532754.

**Funding:** This project was supported by the Australian Sports Commission and Australian

Government through a 2022 Australian Institute of Sport Research Grant. The funders had no role in study design, data collection and analysis, decision to publish, or preparation of the manuscript.

**Competing interests:** The authors have declared that no competing interests exist.

## Conclusion

The development of bench press talent identification and sport transfer models will require future studies to investigate both non-training and training modifiable features, across multiple domains. Large longitudinal studies using information from athlete monitoring databases and multivariable model approaches are needed to understand the interacting features associated with bench press performance, and for the development of talent identification models.

## Introduction

The bench press is one of three lifts in non-disabled powerlifting and the sole lift in Paralympic powerlifting. Bench press is also a popular exercise in individual and team sport strength training programs [1, 2]. To perform the exercise, athletes lay supine on a flat bench and lower a barbell to bottom of their sternum, momentarily hold the bar motionless, before raising the barbell to full elbow extension.

The World Powerlifting Championships are held annually but powerlifting is not an Olympic event. Paralympic Powerlifting on the other hand has been a full medal event since 1984 for men and 2000 for women. During competition, athletes aim to bench press as much weight as possible, while adhering to competition regulations [3, 4].

Key determinants of bench press performance have been investigated in non-disabled athletes [5–8], and more recently, in Para athlete populations [9]. For example, in both non-disabled and Para athlete populations, studies have investigated the association between anthropometric [6–8, 10], body composition [10–12], and technical [5, 13–15] features, and bench press one-repetition maximum (1RM). Understanding the factors that underpin maximum bench press performance could be useful for identifying talented young athletes for selection in state or national powerlifting training programs, similar to the approach used in Olympic weightlifting [16]. Many sports use talent identification models encompassing a range of anthropometric, physical performance and cognitive variables [17] but no uniform approach is currently available for in powerlifting. Information on important features of bench press performance could also be used to identify Para athletes who are good candidates for sport transfer into Paralympic powerlifting. This may be particularly relevant to older Para athletes in power-based sports, as the age-related decline in muscular power is faster than the decline in muscular strength [18]. Para athletes often report difficulty transitioning into retirement [19] and prolonging their career by transferring into a sport with an older age of peak performance may be beneficial for para athlete well-being.

Despite the popularity of the bench press exercise, and its use in non-disabled powerlifting and Paralympic powerlifting competitions, no previous study has synthesized literature studying features of bench press performance. The aim of this scoping review was to summarise features related to bench press 1RM performance in non-disabled and Para athletes. We chose a scoping review approach as our intention was to summarise a broad range of studies that have investigated features related to bench press strength, and to provide recommendations for future studies.

## Method

The scoping review was conducted according to the methods outlined by Arksey and O'Malley [20]. Reporting was guided by the Preferred Reporting Items for Systematic Reviews and Meta-analyses (PRISMA) extension for Scoping Reviews [21]. Ethical approval for this study is not required as it is a systematic review using publicly available data. A protocol for the review was preregistered on the Open Science Framework and is available at https://doi.org/10.17605/OSF.IO/Z52XY.

### Search strategy and selection criteria

The search strategy combined indexed terms with keywords for bench press performance in non-disabled and Para athlete populations. No date restrictions were placed on the search strategy. The search strategy included (but was not limited to) the keywords "resistance training", "para powerlifting", "bench press", "one repetition max" and "strength" with combinations of Boolean operators. Specific search strings used to search databases are provided in S1 File. Searches were completed in the PubMed, SportDiscus and EMBASE databases. The search was conducted on September 8, 2023.

To be included studies needed to be peer-reviewed research articles that reported findings on demographic, anthropometric, body composition, biomechanical, or neuromuscular features related to bench press 1RM. Observational and experimental study designs were included. Studies were included if participants had at least six-months experience in performing the bench press exercise, and if participants were able to bench press a weight equivalent to, or greater than, their body mass. Only studies written in English were included.

We excluded studies that: 1) focused on bench throw or isometric bench press performance, which are distinctly different to the traditional bench press lift; 2) investigated the influence of supplements or ergogenic aids on bench press performance; 3) included non-disabled and Para athletes, but did not separately report the results for each population were excluded; and 4) included strength-based predictor variables, for example, 1RM pull up strength. Conference proceedings, study protocols, letters to the editor, commentaries, and systematic or scoping reviews were excluded.

Title and abstract screening were completed using Rayyan [22]. Two authors (RLB, DNB) determined the eligibility of each record for full-text review, with disagreements resolved by discussion. Full text reviews were completed by authors RLB and DNB. At the full-text review stage we hand-searched the reference lists of studies to identify further studies for inclusion.

### Charting the data

Data collection involved the extraction of general study information (i.e., author names, year of publication) in addition to details about the study objective, design and sample, the data analysis methods, and the key study findings. We recorded whether each study was pre-registered, whether a dataset was publicly shared, and whether study reporting was guided by a checklist, for example, the Strengthening the Reporting of Observational Studies in Epidemiology (STROBE) Statement.

We extracted the study sample size, the number of female participants, the sample age, body mass, height, and absolute and relative bench press 1RM. We also extracted whether or not supportive equipment (e.g. bench shirt or wrist straps) was used during 1RM attempts. Where studies did not state that supportive equipment was used, we assumed it was not. Additionally, we assumed that supportive equipment was used in studies analysing competition data. When correlations were reported, we extracted the type of correlation used (e.g., Pearson, Spearman), the correlation coefficient value and the associated $p$-value and confidence

interval, and level of confidence. We also noted whether studies visualised the relationship between the feature of interest and bench press 1RM using a scatter plot. For studies using regression analysis, we noted the response distribution used (e.g., Gaussian), and extracted the coefficient of determination ($R^2$) value, or an adjusted $R^2$ value. From studies using between group comparisons, we extracted information about the groups being compared, the mean difference in 1RM between groups and the associated confidence interval, and the level of confidence.

Data were independently extracted (RLB, DNB, JLO) and entered onto a data charting form (Excel spreadsheet). Data extraction checks were undertaken by a second, independent author to the original extraction.

### Collating, summarising and reporting the results

Narrative synthesis was used to describe important features (variables) associated with 1RM bench press performance. Features of interest were extracted from included studies and were grouped into six domains, according to whether the feature was an anthropometric, body composition, demographic, technical, neuromuscular or disability variable. Once organized into the six domains, we collated the number of studies that: 1) examined the correlation between specific variables and bench press 1RM; 2) used regression models to investigate the association between a set of variables and bench press 1RM; and 3) compared bench press 1RM between groups. Results from the narrative synthesis were used to develop a set of recommendations to guide future research.

Data summaries were generated in R [23] using packages from the *tidyverse* [24], and are presented as count (percentage) or median (1st and 3rd quartile), unless otherwise stated. We calculated the proportion of female participants included in studies. Confidence intervals on proportions were calculated using the Clopper-Pearson method for the binomial distribution [25] via the R package *binom* [26].

Most studies used correlation analysis to investigate whether there was an association between feature of interest *x* and bench press 1RM. Ranges of correlation coefficients (point estimates) are presented in tabular form, according to each feature studied. We calculated 95% confidence intervals (CIs) on correlation coefficients using the Fisher *z* method [27] via the R package *psychometric* [28]. While no formal analyses were undertaken to aggregate correlation effect sizes, associations were considered positive, negative, or unclear, based on 95% CIs. Features were deemed to have a positive or negative association, depending on the direction of the effect, with bench press 1RM when the 95% CI of both the smallest and largest correlation coefficient excluded the null value of zero. Associations were deemed unclear when the 95% CI of the smallest or largest correlation coefficient included zero. In interpreting the results, we primarily focused on features that had been investigated by at least two studies.

The PRISMA flowchart was produced using the *PRISMA2020* Shiny application [29]. All other figures were generated using the R package *ggplot2* [30]. The data and R code are located in the 'para_power_review' repository available at 10.5281/zenodo.10532751.

## Results

### Search strategy to identify relevant studies

Our search strategy returned 3,999 records from three databases (Fig 1). We removed 1,488 duplicate records before title and abstract screening. Title and abstract screening identified 113 articles for full-text review. The three main reasons for exclusion after full-text review were 1) inclusion of features that were not an anthropometric, body composition, technical, neuromuscular or disability related variable; 2) inclusion of participants with less than 6

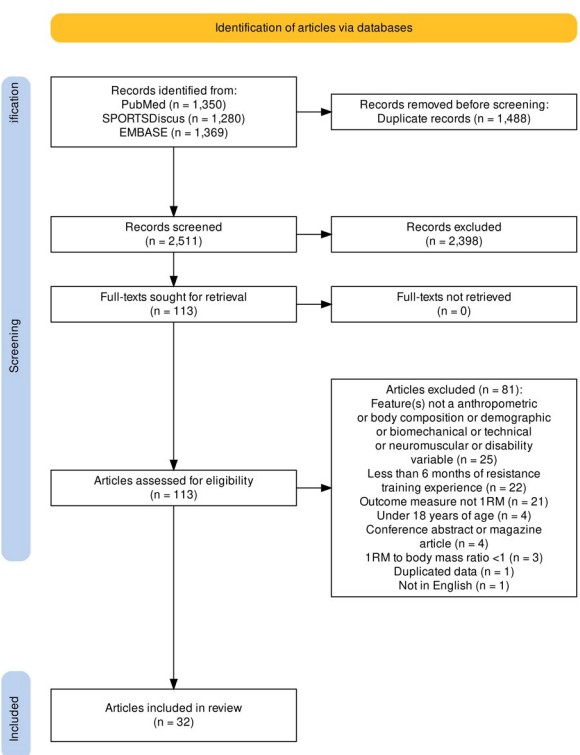

**Fig 1. The PRISMA flow diagram detailing the databases searched and records retrieved, the number of records screened, the full texts retrieved, and the number of studies included in the scoping review.**

months of resistance training experience, 3) not using 1RM as the outcome measure (Fig 1). Data extraction was completed for 32 studies.

We included male but excluded female athletes from three studies [7, 15, 31], as the female athlete groups were not able to bench press their body mass, and therefore, did not meet our inclusion criteria. Similarly, only the "high strength" group from one study [32] was included, as the "low strength" group were not able to bench press their body mass. Results were taken from the cross-validation set in the study by Hetzler and Colleagues [11].

## Study population

**Non-disabled athlete studies.** The 24 studies on non-disabled athletes involved 2,686 individuals (Fig 2A). Of the 2,686 athletes, 21.9% were female (Fig 2B). All 24 studies involved males. Six studies involved females but female groups from only three studies met the inclusion criteria and were therefore included in the review (Table 1). The median number of participants per study was 30 (1st and 3rd quartile = 20 to 68). The two most common studied populations were college athletes or university students, and powerlifters (Table 1). Ten studies (41.7%) involved college athletes or university students with a background in resistance training, and seven studies (29.2%) recruited sub-elite, national or international level powerlifters.

For non-disabled athlete male groups, the median (study sample average) age, body mass, and height was 22.5 years (1st and 3rd quartile = 21.1 to 24.9; Fig 2C), 86.8 kg (1st and 3rd quartile = 78.3 to 93.7), and 1.78 m (1st and 3rd quartile = 1.76 to 1.80). The median 1RM and 1RM to body mass ratio was 124.2 kg (1st and 3rd quartile = 94.2 to 145.7; Fig 2D) and 1.30 (1st and 3rd quartile = 1.21 to 1.57; Fig 2E).

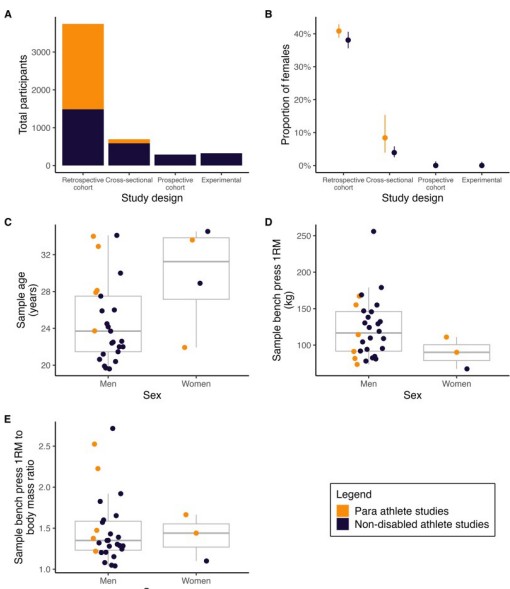

**Fig 2.** The total number of participants (panel A) and proportion of female participants (panel B) according to study design, and average age (panel C), bench press 1RM (panel D) and bench press 1RM to body mass ratio (panel E) of study samples.

Only one non-disabled athlete study involving females [6] reported complete age, body mass and height information for the female sample (Table 1). However, 1RM data were not reported separately for males and females.

**Para athlete studies.** The eight studies involving Para athlete populations included a total of 2,364 individuals (Fig 2A). Of these, 39.4% were female (Fig 2B). All eight studies involved males; four studies involved females, and it was unclear whether one study involved females (Table 1). The median number of participants per study was 30 (1st and 3rd quartile = 15 to 121). International and national Paralympic powerlifters were the most common populations studied, with two studies (25%) involving national level wheelchair basketball players (Table 1).

Four [10, 13, 33, 34] of eight studies (50%) involving Para athletes reported participant impairment type, three [33–35] of the eight studies (37.5%) reported origin of injury (i.e., congenital or acquired), and five [13, 14, 34–36] of the eight studies (62.5%) reported sport classification information. The three studies [10, 13, 33] that did not use competition data and reported impairment type recruited a total of 80 participants. In this group of 80 Para athletes, the most common impairment types were spinal cord injury (n = 30/80; 37.5%) and amputation (n = 21/80; 26.3%).

For male Para athlete groups, the median (study sample average) age and body mass 28.1 years (1st and 3rd quartile = 27.9 to 32.9; Fig 2C) and 67.7 kg (1st and 3rd quartile = 60.0 to 75.7). The median 1RM and 1RM to body mass ratio was 102.8 kg (1st and 3rd quartile = 84.1 to 144.9; Fig 2D) and 1.47 (1st and 3rd quartile = 1.38 to 2.23; Fig 2E).

Three studies reported age and body mass for female Para athlete groups but only two studies reported absolute and relative 1RM (Table 1). The average sample age of female groups ranged from 21.9 to 33.6 years. The average sample body mass of female groups ranged from 62.6 to 66.7 kg. Absolute and relative 1RM for two female groups are reported in Table 1.

**Table 1.  Summary of non-disabled (n = 24) and para (n = 8) athlete studies.**

| First author | Year | Population | Sample size, n | Women, n (%) | Age, years | Body mass, kg | Height, m | 1RM, kg | 1RM to body mass ratio |
|---|---|---|---|---|---|---|---|---|---|
| *Non-disabled athlete studies* | | | | | | | | | |
| *Cross-sectional studies* | | | | | | | | | |
| McLaughlin [50] | 1984 | National and international powerlifters | 45 | 0 (0) | NR | 94.2 | NR | 255.9 | 2.72 |
| Wagner [46] | 1992 | College students | 24 | 0 (0) | 21.5 | 85.3 | 1.78 | 118.9 | 1.39 |
| Brechue [12] | 2002 | National powerlifters | 20 | 0 (0) | NR | 92.5 | 1.69 | 168.9 | 1.83 |
| Hetzler [11] | 2010 | Division 1 college football players | 118 | 0 (0) | 19.7 | 103.1 | 1.84 | 138.3 | 1.35 |
| Caruso [8] | 2012 | College students and athletes | 36 | 0 (0) | NR | 86.8 | NR | 104.4 | 1.20 |
| Winwood [45] | 2012 | Semi-professional rugby players | 23 | 0 (0) | 22.0 | 102.6 | 1.85 | 132.2 | 1.29 |
| Ye [47] | 2013 | National powerlifters | 20 | 0 (0) | 30.0 | 93.2 | 1.69 | 179.0 | 1.92 |
| Akagi [39] | 2014 | College athletes | 18 | 0 (0) | 20.6 | 69.6 | 1.72 | 91.9 | 1.32 |
| Kerksick [32] | 2014 | Recreationally trained | 66 | 0 (0) | 24.5 | 91.9 | 1.79 | 124.2 | 1.35 |
| Schumacher [43] | 2016 | Division 2 college footballers | 34 | 0 (0) | 20.4 | 99.1 | 1.82 | 129.2 | 1.30 |
| Loturco [49] | 2017 | Rugby and combat sport athletes | 36 | 0 (0) | 24.2 | 86.8 | 1.81 | 109.7 | 1.28 |
| Ferland [7] | 2020 | Sub-elite powerlifters | 9 | 0 (0)* | 27.5 | 93.5 | 1.76 | 146.9 | 1.57 |
| Reya [5] | 2021 | Competitive powerlifters | 13 | 0 (0) | 26.0 | 93.8 | 1.78 | 155.0 | 1.65 |
| Ferrari [6] | 2022 | National powerlifters | 74 | 23 (31) | M: 25.9 W: 28.9 | M: 84.0 W: 63.0 | M: 1.74 W: 1.62 | 130.4 | 1.60 |
| Massini [41] | 2022 | Well trained | 30 | 0 (0) | 23.7 | 78.7 | 1.79 | 82.5 | 1.05 |
| Zaras [44] | 2023 | Physical education and sports science students | 21 | 0 (0) | 22.6 | 76.6 | 1.79 | 95.4 | 1.25 |
| *Experimental studies* | | | | | | | | | |
| Nacleiro [31] | 2017 | Resistance trained athletes | 242 | 0 (0)* | 22.4 | 73.3 | 1.75 | 84.5 | 1.15 |
| Garcia-Ramos [48] | 2018 | College students | 30 | 0 (0) | 21.2 | 72.3 | 1.78 | 78.1 | 1.08 |
| Perez-Castilla [42] | 2020 | Sports science students | 20 | 0 (0) | 22.5 | 77.9 | 1.78 | 81.0 | 1.04 |
| Garcia-Ramos [15] | 2021 | College students | 12 | 0 (0)* | 19.9 | 78.1 | 1.80 | 94.2 | 1.21 |
| Lee [51] | 2023 | Recreationally trained lifters | 20 | 0 (0) | 22.0 | 84.9 | 1.77 | 109.0 | 1.28 |
| *Prospective cohort studies* | | | | | | | | | |
| Mann [40] | 2012 | Division 1 college athletes | 289 | 0 (0) | 19.6 | 103.5 | 1.86 | 145.7 | 1.43 |
| *Retrospective cohort studies* | | | | | | | | | |
| Latella [37]# | 2018 | International competitive powerlifters | 1368 | 518 (38) | 23–78† | R‡ | NR | R | R |
| Solberg [38]# | 2019 | International powerlifters | 118 | 48 (41) | M: 34.1 W: 34.5 | R‡ | NR | R | NR |
| *Para athlete studies* | | | | | | | | | |
| *Cross-sectional* | | | | | | | | | |
| Loturco [52] | 2019 | National Paralympic powerlifters | 17 | NR | NR | NR | NA | 131.8 | 1.92 |
| Hamid [10] | 2019 | National and state Paralympic powerlifters | 52 | 9 (17) | M: 23.0§ W: 28.0§ | M: 66.6§ W: 66.1§ | NA | M: 91.3 W: NR | M: 1.38 W: NR |
| Iturricastillo [13] | 2019 | National wheelchair basketballers | 9 | 0 (0) | 34.0 | NR | NA | 81.7 | NR |
| Teles [33] | 2021 | Competitive Paralympic powerlifters | 19 | 0 (0) | 28.1 | 77.59 | NA | 114.4 | 1.47 |
| Romarate [14] | 2021 | National wheelchair basketballers | 10 | 0 (0) | 27.9 | 60.3 | NA | 73.5 | 1.22 |
| *Retrospective cohort* | | | | | | | NA | | |
| Lopes-Silva [35]# | 2022 | Paralympic powerlifters (world record holders) | 40 | 20 (50) | M: 23.7 W: 21.9 | M: 58.9 W: 66.7 | NA | M: 155.1 W: 111.0 | M: 2.53 W: 1.66 |

*(Continued)*

**Table 1.** (Continued)

| First author | Year | Population | Sample size, n | Women, n (%) | Age, years | Body mass, kg | Height, m | 1RM, kg | 1RM to body mass ratio |
|---|---|---|---|---|---|---|---|---|---|
| Severin [36]# | 2023 | International Paralympic powerlifters | 328 | 157 (48) | R | R‡ | NA | R | NR |
| Lopes-Silva [34]# | 2023 | Paralympic powerlifters | 1889 | 745 (39) | M: 32.9 W: 33.6 | M: 75.1 W: 62.6 | NA | M: 167.2 W: 90.1 | M: 2.23 W: 1.44 |

Note. Age, body mass, height, 1RM and 1RM to body mass ratio values are the sample or group mean, unless otherwise indicated. Height was considered not applicable for studies involving Para athlete populations.

M = Men, NA = Not applicable, NR = Not reported, R = Longitudinally reported, W = Women, 1RM = One repetition maximum.

\# Study used longitudinal competition data

\* Female group from the study excluded because the group mean 1RM to body mass ratio was less than 1.

† Range.

‡ Competition weight categories.

§ Group median.

## Study design

**Non-disabled athlete studies.** In non-disabled athlete populations, cross-sectional study designs were most common (16/24; 66.7%), followed by experimental studies (5/24; 20.1%), retrospective cohort studies (2/24; 8.3%), and prospective cohort studies (1/24; 4.2%; Table 1). Two studies [37, 38] used published competition results.

**Para athlete studies.** Cross-sectional study designs (5/8; 62.5%) were most common in Para athlete populations, followed by retrospective cohort studies (3/8; 12.5%; Table 1). Three studies [34–36] involving Para athletes used published competition results.

## Features related to bench press one repetition maximum performance

Fig 3A provides a graphical summary of the number of studies that have explored anthropometric, body composition, demographic, technical, neuromuscular and disability features. Fig 3B shows the number of studies that investigated specific feature combinations. Supportive equipment was used during 1RM assessments in 11 studies, including four studies involving Para-athletes and seven studies involving non-disabled athletes.

**Non-disabled athlete studies.** Anthropometric features were investigated by 16 (66.7%) studies involving non-disabled athletes [6–8, 11, 12, 32, 37, 39–46]. Thirteen studies used correlation analysis to investigate 36 unique anthropometric features (Table 2). Of the features studied more than once, body mass, triceps muscle thickness, flexed upper arm circumference, arm cross-sectional area, forearm circumference, and calf and thigh circumference were positively associated with 1RM bench press (Table 2). The association between height, triceps fascicle length, triceps pennation angle, arm length, chest circumference, hip circumference, and bench press 1RM was unclear (Table 2). Using a between group design, two studies [37, 38] investigated the association between relative or absolute bench press strength and competition weight classes. Latella et al. [37] found that relative strength was generally higher in the lightest weight classes compared to the heaviest weight classes, with effect sizes ranging from a Cohens $d$ of 0.73 (95% CI = 0.39 to 1.07) to 1.11 (95% CI = 0.71 to 1.50). Solberg et al. [38] found that absolute bench press strength increased in a hierarchical manner with weight categories, in elite male and female powerlifters. Three studies included anthropometric features in a regression model [8, 32, 42]. Included features were arm length [8, 42], biacromial width [8], and height [32].

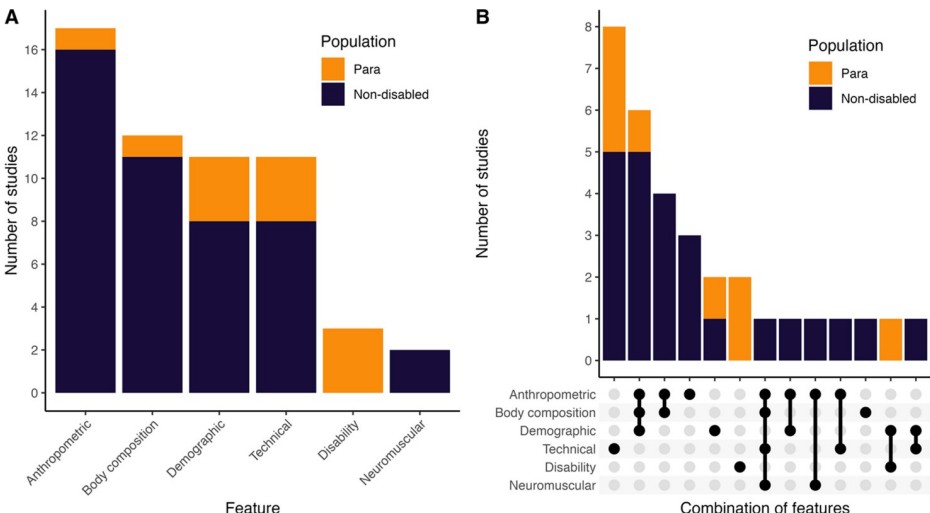

**Fig 3.** The number of studies investigating anthropometric, body composition, technical, disability or neuromuscular features of bench press one-repetition maximum performance (panel A), and the number of studies that investigated unique feature combinations (panel B).

Body composition features were investigated by 11 (45.8%) studies involving non-disabled athlete populations [6, 7, 11, 12, 32, 40, 41, 43, 45, 47]. All 11 studies used correlation analysis to investigate a total of 48 unique features (Table 3). Only whole-body features had been investigated by more than one study, with lean body mass, body mass index and skeletal muscle mass positively associated with bench press 1RM (Table 3). The association between fat mass and 1RM and between body fat percentage and 1RM was unclear (Table 3). No studies investigated body composition features in a non-disabled athlete population using a between group design. Three studies included body composition features in a regression model [8, 32, 41]. Included features were: body mass [8], fat free mass [32, 41], fat mass [32], appendicular fat free mass index [41], upper limb fat free mass [41], and arm cross sectional area [41].

Eight (33.3%) studies involving non-disabled athlete populations investigated demographic features, being age [6, 7, 11, 32, 38, 40] or sex [31, 37]. Five studies used correlation analysis to investigate the association between age and bench press 1RM (Table 4). The association between age and bench press 1RM was unclear (Table 4). Two studies investigated differences in demographic variables using a between group design [37, 38]. Latella et al. [37] found that relative 1RM bench press strength was on average higher in males compared to females (Cohen's $d$ = 1.84, 95% CI = 1.71 to 1.97). Relative bench press 1RM was also higher in the open age competition (23 to 39 years) compared to master's categories (≥40 years), with the standardised difference between the Open age group and Masters groups ranging from medium ($d$ = 0.58, 95% CI = 0.35 to 0.81) to large ($d$ = 1.33, 95% CI = 0.94 to 1.71). Solberg et al. [38] found that bench press performance continues to improve until mid-to-late 30's ties, in elite male and female powerlifters. One study included sex in a regression model [31]. Regression analysis results are summarised below.

Technical features were investigated by eight (33.3%) studies involving non-disabled athlete populations [5, 15, 31, 42, 48–51]. Two studies used correlation analysis to investigate a total of nine unique technical features, with no individual feature investigated by more than a single study (Table 4). Two studies examined differences in technical features using a between group design, with both studies reporting that optimum bench press 1RM occurs at about 200% of

**Table 2. Summary of correlations between anthropometric variables and 1RM bench press for non-disabled athletes.**

| Variable | Studies, n | Corr's, n | Correlation coefficient (95% CI) | Association |
|---|---|---|---|---|
| *Whole body* | | | | |
| Body mass [6–8, 11, 32, 40, 45] | 7 | 7 | 0.16 (0.05, 0.27) to 0.80 (0.29, 0.96) | Positive |
| Height [6, 7, 11, 32, 40, 45] | 6 | 6 | −0.43 (−0.85, 0.33) to 0.53 (0.44, 0.61) | Unclear |
| *Upper limb* | | | | |
| Upper arm circumference flexed [5, 6, 11, 43, 45] | 5 | 6 | 0.58 (0.04, 0.86) to 0.90 (0.85, 0.94) | Positive |
| Arm cross sectional area [11, 41, 43] | 3 | 3 | 0.68 (0.44, 0.83) to 0.98 (0.97, 0.99) | Positive |
| Arm length [6, 8, 11] | 3 | 3 | −0.07 (−0.29, 0.16) to 0.50 (0.35, 0.62) | Unclear |
| Triceps muscle thickness [12, 44] | 2 | 2 | 0.53 (0.13, 0.78) to 0.81 (0.57, 0.92) | Positive |
| Forearm circumference [5, 6] | 2 | 2 | 0.26 (0.03, 0.46) to 0.70 (0.24, 0.90) | Positive |
| Triceps fascicle length [5, 44] | 2 | 2 | 0.16 (−0.29, 0.55) to 0.27 (−0.33, 0.71) | Unclear |
| Triceps pennation angle [5, 44] | 2 | 2 | −0.46 (−0.81, 0.12) to 0.41 (−0.03, 0.72) | Unclear |
| Upper arm circumference relaxed [6] | 1 | 1 | 0.87 (0.80, 0.92) | Positive |
| Forearm muscle thickness [12] | 1 | 1 | 0.82 (0.59, 0.93) | Positive |
| Wrist circumference [6] | 1 | 1 | 0.78 (0.67, 0.86) | Positive |
| Biceps muscle thickness [12] | 1 | 1 | 0.77 (0.50, 0.90) | Positive |
| Forearm to upperarm length ratio [5] | 1 | 1 | 0.60 (0.07, 0.87) | Positive |
| Triceps cross-sectional area [5] | 1 | 1 | 0.58 (0.04, 0.86) | Positive |
| Arm length to height ratio [5] | 1 | 1 | −0.31 (−0.74, 0.29) | Unclear |
| Arm span [5] | 1 | 1 | −0.02 (−0.56, 0.54) | Unclear |
| *Trunk* | | | | |
| Chest circumference [5, 6, 43, 45] | 4 | 4 | 0.15 (−0.44, 0.65) to 0.67 (0.52, 0.78) | Unclear |
| Hip circumference [6, 45] | 2 | 2 | 0.22 (−0.21, 0.58) to 0.35 (0.13, 0.54) | Unclear |
| Pectoralis major cross-sectional area [39] | 1 | 1 | 0.86 (0.66, 0.95) | Positive |
| Subscapula muscle thickness [12] | 1 | 1 | 0.85 (0.65, 0.94) | Positive |
| Neck circumference [6] | 1 | 1 | 0.79 (0.69, 0.86) | Positive |
| Waist circumference [6] | 1 | 1 | 0.79 (0.69, 0.86) | Positive |
| Chest muscle thickness [12] | 1 | 1 | 0.77 (0.50, 0.90) | Positive |
| Biacromial width [8] | 1 | 1 | 0.34 (0.01, 0.60) | Positive |
| Abdomen muscle thickness [12] | 1 | 1 | 0.35 (−0.11, 0.69) | Unclear |
| Chest circumference to height ratio [5] | 1 | 1 | 0.32 (−0.28, 0.74) | Unclear |
| Chest depth [5] | 1 | 1 | 0.15 (−0.44, 0.65) | Unclear |
| Iliac to acromial width ratio [5] | 1 | 1 | −0.20 (−0.68, 0.39) | Unclear |
| *Lower limb* | | | | |
| Calf circumference [6, 45] | 2 | 3 | 0.43 (0.22, 0.60) to 0.67 (0.36, 0.85) | Positive |
| Thigh circumference [6, 45] | 2 | 3 | 0.25 (0.02, 0.45) to 0.51 (0.32, 0.66) | Positive |
| Tibialis anterior muscle thickness [12] | 1 | 1 | 0.82 (0.59, 0.93) | Positive |
| Calf muscle thickness [12] | 1 | 1 | 0.78 (0.52, 0.91) | Positive |
| Hamstrings muscle thickness [12] | 1 | 1 | 0.69 (0.36, 0.87) | Positive |
| Quadriceps muscle thickness [12] | 1 | 1 | 0.67 (0.32, 0.86) | Positive |
| Ankle circumference [6] | 1 | 1 | 0.54 (0.36, 0.68) | Positive |

Note. Features are arranged by number of studies, number of correlations, association, and correlation coefficient.

CI = Confidence interval.

the biacromial distance [46, 51]. Four studies included technical features in a regression model [15, 31, 42, 49], namely, mean barbell velocity [15, 31, 42] and propulsive barbell velocity [49].

Neuromuscular features were investigated by two (8.3%) studies involving non-disabled athlete populations [5, 44]. Both studies used correlation analysis to study a total of six unique

**Table 3. Summary of correlations between body composition variables and 1RM bench press for non-disabled athletes.**

| Variable | Studies, n | Corr's, n | Correlation coefficient (95% CI) | Association |
|---|---|---|---|---|
| *Whole body* | | | | |
| Lean body mass [5–7, 11, 12, 32, 41, 43, 45] | 9 | 11 | 0.49 (0.18, 0.71) to 0.88 (0.72, 0.95) | Positive |
| Body fat percentage [6, 7, 11, 32, 43, 45] | 6 | 6 | −0.57 (−0.71, −0.39) to 0.62 (−0.08, 0.91) | Unclear |
| Fat mass [5, 7, 32] | 3 | 4 | −0.12 (−0.63, 0.46) to 0.62 (−0.08, 0.91) | Unclear |
| Lean body mass to height ratio [6, 12] | 2 | 2 | 0.87 (0.70, 0.95) to 0.88 (0.82, 0.92) | Positive |
| Skeletal muscle mass [45, 47] | 2 | 2 | 0.59 (0.23, 0.81) to 0.88 (0.72, 0.95) | Positive |
| Body mass index [7, 40] | 2 | 2 | 0.58 (0.50, 0.65) to 0.79 (0.26, 0.95) | Positive |
| Skeletal muscle mass to height ratio [47] | 1 | 1 | 0.87 (0.70, 0.95) | Positive |
| Bone mineral content total [7] | 1 | 1 | 0.87 (0.65, 0.96) | Positive |
| Percentage lean body tissue [7] | 1 | 1 | 0.86 (0.46, 0.97) | Positive |
| Appendicular lean mass [41] | 1 | 1 | 0.83 (0.67, 0.92) | Positive |
| Muscle thickness sum of 13 sites [12] | 1 | 1 | 0.81 (0.57, 0.92) | Positive |
| Bone mass [5] | 1 | 1 | 0.65 (0.15, 0.88) | Positive |
| Bone mineral density total [7] | 1 | 1 | 0.59 (0.11, 0.85) | Positive |
| Sum of skinfolds [43] | 1 | 1 | 0.46 (0.14, 0.69) | Positive |
| *Upper limb* | | | | |
| Upper limb fat free mass [41] | 1 | 1 | 0.91 (0.82, 0.96) | Positive |
| Muscle thickness sum of upper body sites [12] | 1 | 1 | 0.85 (0.65, 0.94) | Positive |
| Bone mineral content arms [7] | 1 | 1 | 0.83 (0.55, 0.94) | Positive |
| Arms total mass [7] | 1 | 1 | 0.77 (0.22, 0.95) | Positive |
| Lean mass arms [7] | 1 | 1 | 0.72 (0.11, 0.94) | Positive |
| Upper limb fat mass [7] | 1 | 1 | 0.61 (−0.09, 0.91) | Unclear |
| Upper limb boy fat percentage [7] | 1 | 1 | 0.53 (−0.21, 0.88) | Unclear |
| Bone mineral density arms [7] | 1 | 1 | 0.43 (−0.11, 0.77) | Unclear |
| Triceps skinfold [43] | 1 | 1 | 0.31 (−0.03, 0.59) | Unclear |
| *Trunk* | | | | |
| Bone mineral density ribs [7] | 1 | 1 | 0.91 (0.75, 0.97) | Positive |
| Bone mineral content ribs [7] | 1 | 1 | 0.85 (0.60, 0.95) | Positive |
| Bone mineral content trunk [7] | 1 | 1 | 0.85 (0.60, 0.95) | Positive |
| Trunk total mass [7] | 1 | 1 | 0.83 (0.37, 0.96) | Positive |
| Lean mass trunk [7] | 1 | 1 | 0.82 (0.34, 0.96) | Positive |
| Bone mineral content pelvis [7] | 1 | 1 | 0.81 (0.51, 0.93) | Positive |
| Bone mineral content spine [7] | 1 | 1 | 0.79 (0.47, 0.93) | Positive |
| Bone mineral density trunk [7] | 1 | 1 | 0.79 (0.47, 0.93) | Positive |
| Bone mineral density spine [7] | 1 | 1 | 0.62 (0.16, 0.86) | Positive |
| Bone mineral density pelvis [7] | 1 | 1 | 0.53 (0.02, 0.82) | Positive |
| Subscapula skinfold [43] | 1 | 1 | 0.47 (0.16, 0.70) | Positive |
| Abdominal skinfold [43] | 1 | 1 | 0.43 (0.11, 0.67) | Positive |
| Fat mass trunk [7] | 1 | 1 | 0.65 (−0.02, 0.92) | Unclear |
| Trunk body fat percentage [7] | 1 | 1 | 0.49 (−0.26, 0.87) | Unclear |
| Bone mineral content head [7] | 1 | 1 | 0.38 (−0.16, 0.75) | Unclear |
| Bone mineral density head [7] | 1 | 1 | 0.06 (−0.47, 0.56) | Unclear |
| *Lower limb* | | | | |
| Muscle thickness sum of lower body sites [12] | 1 | 1 | 0.79 (0.53, 0.91) | Positive |
| Bone mineral content legs [7] | 1 | 1 | 0.79 (0.47, 0.93) | Positive |
| Legs total mass [7] | 1 | 1 | 0.72 (0.11, 0.94) | Positive |
| Bone mineral density legs [7] | 1 | 1 | 0.57 (0.08, 0.84) | Positive |

*(Continued)*

**Table 3.** (Continued)

| Variable | Studies, n | Corr's, n | Correlation coefficient (95% CI) | Association |
|---|---|---|---|---|
| Lower limb fat mass [7] | 1 | 1 | 0.50 (−0.25, 0.87) | Unclear |
| Lean mass legs [7] | 1 | 1 | 0.47 (−0.28, 0.86) | Unclear |
| Lower limb body fat percentage [7] | 1 | 1 | 0.38 (−0.38, 0.83) | Unclear |

Note. Features are arranged by number of studies, number of correlations, association, and correlation coefficient.

CI = Confidence interval.

features (Table 4). No feature was investigated by more than one study, and only shoulder press rate of force development was found to be positively associated with bench press 1RM (Table 4). No studies included neuromuscular features in a regression model.

Seven non-disabled athlete studies reported regression models [8, 15, 31, 32, 41, 42, 49]. Several models included barbell velocity (peak or average) during a submaximal contraction as a univariate predictor [15, 42, 49]. These models explained 95% to 96% of the variance in bench press 1RM. Four studies fitted regression models with more than one predictor variable [8, 31, 32, 41]. In resistance trained male athletes, barbell velocity and sex explained 84% of the variance in bench press 1RM [31]. In recreationally trained males, fat mass, fat free mass and height explained 82% of the variance in bench press 1RM [32]. In well trained males, appendicular fat free mass index, whole body fat free mass, upper limb fat free mass and arm cross sectional area explained 82% of the variance in bench press 1RM [41]. In male college students and athletes, body mass, arm length and biacromial width explained 64% of the variance in bench press 1RM [8].

**Table 4. Summary of correlations between demographic, technical and neuromuscular variables and 1RM bench press for non-disabled athletes.**

| Variable | Studies, n | Corr's, n | Correlation coefficient (95% CI) | Association |
|---|---|---|---|---|
| *Demographic* | | | | |
| Age [6, 7, 11, 32, 40] | 5 | 5 | −0.18 (−0.39, 0.05) to 0.68 (0.03, 0.93) | Unclear |
| *Technical* | | | | |
| Load-velocity relationship [48] | 1 | 2 | 0.99 (0.98, 1) to 1 (1, 1) | Positive |
| Shoulder flexion angle at sticking point [5] | 1 | 1 | 0.49 (−0.08, 0.82) | Unclear |
| Shoulder abduction angle at sticking point [5] | 1 | 1 | 0.47 (−0.11, 0.81) | Unclear |
| Horizontal shoulder adduction angle sticking point [5] | 1 | 1 | 0.25 (−0.35, 0.70) | Unclear |
| Horizontal bar displacement [5] | 1 | 1 | 0.24 (−0.36, 0.70) | Unclear |
| Lumbar arch height [5] | 1 | 1 | 0.2 (−0.39, 0.68) | Unclear |
| Foot ground reaction force during bench press [5] | 1 | 1 | 0.12 (−0.46, 0.63) | Unclear |
| Elbow flexion angle at sticking point [5] | 1 | 1 | −0.4 (−0.78, 0.19) | Unclear |
| Vertical bar displacement [5] | 1 | 1 | −0.31 (−0.74, 0.29) | Unclear |
| *Neuromuscular* | | | | |
| Shoulder press rate of force development [44] | 1 | 7 | 0.49 (0.07, 0.76) to 0.72 (0.42, 0.88) | Positive |
| Deltoid EMG during bench press [5] | 1 | 1 | 0.22 (−0.38, 0.69) | Unclear |
| Triceps voluntary activation [5] | 1 | 1 | 0.18 (−0.41, 0.67) | Unclear |
| Latissimus dorsi EMG during bench press [5] | 1 | 1 | −0.26 (−0.71, 0.34) | Unclear |
| Triceps EMG during bench press [5] | 1 | 1 | −0.17 (−0.66, 0.42) | Unclear |
| Pectoralis major EMG during bench press [5] | 1 | 1 | −0.09 (−0.61, 0.49) | Unclear |

Note. Features are arranged by number of studies, number of correlations, association, and correlation coefficient.

CI = Confidence interval, EMG = Electromyography

**Para athlete studies.** Disability features were investigated by three (37.5%) studies involving Para athletes [33–35]. These three studies used a between group design to investigate the effect of origin of impairment on bench press 1RM. Two studies [33, 34] reported that athletes with an acquired impairment had on average a higher bench press 1RM than those with congenital impairments (Cohen's $d$ = 0.96 [33]; $\eta_p^2$ = 0.01 [34]). In contrast, another study [35] found no difference in absolute or relative bench press 1RM between acquired and congenital impairments ($d$ = –0.39 to 0.64), in males and females. Two separate studies found that athletes with acquired impairments held more Paralympic records (70% vs. 30%) [35] and won more medals (males = 61% vs. 39%; females = 64% vs. 36%) [34] compared to athletes with congenital impairments. Lopes-Silva et al. [34] found that Para athletes with a limb deficiency had a higher number of medals than those with other types of deficiencies, with other deficiencies including leg length differences, impairments in range of movement and muscle power, hypertonia, ataxia, athetosis and short stature.

Anthropometric features were investigated by only one study involving Para athletes [10]. Using correlation analysis, the authors found that of the nine unique anthropometric features studied, body mass, relaxed and flexed arm circumference, and hip circumference were positively associated with bench press 1RM (Table 5). The same study also investigated four unique body composition features, with body mass index and lean body mass positively associated with bench press 1RM (Table 5).

The demographic feature of age was investigated by two Para athlete studies [10, 36]. Using correlation analysis, Hamid et al. [10] found there was a positive association between age and bench press 1RM (Table 5). Using regression analysis, Severin et al. [36] examined the peak age of competition performance for male and female Paralympic powerlifters, considering the

**Table 5. Summary of correlations between disability, anthropometric, body composition, demographic, and technical variables and 1RM bench press for Para athletes.**

| Variable | Studies, n | Corr's, n | Correlation coefficient (95% CI) | Association |
|---|---|---|---|---|
| *Anthropometric* | | | | |
| Upper arm circumference flexed [10] | 1 | 2 | 0.65 (0.46, 0.78) to 0.69 (0.51, 0.81) | Positive |
| Upper arm circumference relaxed [10] | 1 | 2 | 0.54 (0.31, 0.71) to 0.66 (0.47, 0.79) | Positive |
| Arm length [10] | 1 | 2 | 0.04 (−0.24, 0.31) to 0.07 (−0.21, 0.34) | Unclear |
| Forearm length [10] | 1 | 2 | 0.03 (−0.24, 0.30) to 0.12 (−0.16, 0.38) | Unclear |
| Body mass [10] | 1 | 1 | 0.41 (0.15, 0.61) | Positive |
| Hip circumference [10] | 1 | 1 | 0.31 (0.04, 0.54) | Positive |
| Biacromial width [10] | 1 | 1 | 0.01 (−0.26, 0.28) | Unclear |
| Waist circumference [10] | 1 | 1 | 0.01 (−0.26, 0.28) | Unclear |
| Height [10] | 1 | 1 | −0.05 (−0.32, 0.23) | Unclear |
| *Body composition* | | | | |
| Body mass index [10] | 1 | 1 | 0.46 (0.21, 0.65) | Positive |
| Lean body mass [10] | 1 | 1 | 0.39 (0.13, 0.60) | Positive |
| Fat mass [10] | 1 | 1 | 0.24 (−0.04, 0.48) | Unclear |
| Body fat percentage [10] | 1 | 1 | 0.14 (−0.14, 0.40) | Unclear |
| *Demographic* | | | | |
| Age [10] | 1 | 1 | 0.35 (0.09, 0.57) | Positive |
| *Technical* | | | | |
| Barbell mean propulsive velocity [13, 14] | 2 | 4 | −0.94 (−0.99, −0.76) to −0.98 (−0.99, −0.91) | Negative |

Note. Features are arranged by number of studies, number of correlations, association, and correlation coefficient.

CI = Confidence interval.

effect of weight category. Paralympic powerlifters in heavier weight categories were older than those in lower weight categories. Male Paralympic powerlifters achieved their heaviest competition lift at 36 years of age. Females achieved their heaviest lift at 41 years of age. Using a between group design, Lopes-Silva et al. [34] investigated the effect of sex on bench press performance in Para athletes, reporting that bench press 1RM was on average higher in males compared with females.

Technical features were investigated by three (37.5%) studies involving Para athlete populations [13, 14, 52]. Using correlation analysis, two studies [13, 14] found that barbell mean propulsive velocity was negatively associated with bench press 1RM (Table 5). Using regression analysis, Loturco et al. [52] found that peak or mean propulsive barbell velocity during submaximal contractions explained 87% or more of the variation in bench press 1RM in males ($R^2$ = 0.88 to 0.91), females ($R^2$ = 0.87 to 0.90), and a short stature group ($R^2$ = 0.78 to 0.83). In a small group of nine national wheelchair basketballers, propulsive barbell velocity during submaximal contractions explained 94% of the variation in bench press 1RM [13].

No studies involving Para athlete populations investigated the association between neuromuscular features and bench press 1RM.

## Data analysis

**Non-disabled athlete studies.** Twelve studies (50%) used correlation analyses only, four (16.7%) used group difference analyses, another four studies (16.7%) used linear regression, three studies (12.5%) used both correlation and linear regression, and one study [50] used descriptive analyses only (Table 6).

Fifteen studies used correlation analyses. Only one of these 15 studies reported uncertainty on correlations (i.e., 95% Cis), the confidence intervals around reported correlation coefficients are shown in Fig 4. Four of the 15 studies reported *p*-values for correlations, seven reported *p*-values for some correlations, and another four studies did not report *p*-values (Table 1). Of the 19 studies that used correlation analysis or linear regression, 12 studies did not show a scatter plot, four studies displayed scatter plots for some variables, and three studies displayed scatter plots for all variables (Table 1). All studies that used linear regression reported an $R^2$ or adjusted $R^2$ value (Table 6).

Of the four studies that used group difference analyses, two did not report an effect size for the group difference, with the other two studies reporting a mean or standardised mean difference (Table 6). Both studies that reported a group difference effect size reported it with a measure of uncertainty (Table 6).

No studies involving non-disabled athlete populations were pre-registered or publicly shared their data.

**Para athlete studies.** Three studies (37.5%) used group difference analyses, two studies (25%) used linear regression, two studies (25%) used correlation analyses only, and one (12.5%) study used both correlation and linear regression (Table 6). Of the three studies that used correlation analyses, only one study reported uncertainty on correlations (i.e., 90% Cis). The confidence intervals around reported correlation coefficients are shown in Fig 4. All three studies using correlation analyses reported *p*-values for correlations (Table 6). Five studies used correlation or linear regression, with four of these displaying scatter plots for all studied variables (Table 6). All three studies that used linear regression reported an $R^2$ or adjusted $R^2$ value. Of the three studies that used group difference analyses, only one study reported an effect size for the group difference (i.e., Cohen's *d*) but did not report it with a measure of uncertainty (Table 6). No studies involving Para athletes were pre-registered or publicly shared their data.

**Table 6. Summary of data analysis methods.**

| Author | Year | Study design | Sample size, n | Correlation analysis | Correlation uncertainty | Correlation p-value | Scatter plot | Regression analysis | R-squared | Between group analysis | Group difference effect size | Group difference uncertainty |
|---|---|---|---|---|---|---|---|---|---|---|---|---|
| *Non-disabled athlete studies* | | | | | | | | | | | | |
| McLaughlin [50] | 1984 | Cross-sectional | 45 | F | NA | NA | NA | F | NA | F | NA | NA |
| Wagner [46] | 1992 | Cross-sectional | 24 | F | NA | NA | NA | F | NA | T | F | F |
| Brechue [12] | 2002 | Cross-sectional | 20 | Pearson | F | Partial | Partial | F | NA | F | NA | NA |
| Hetzler [11] | 2010 | Cross-sectional | 118 | Pearson | F | Partial | F | F | NA | F | NA | NA |
| Caruso [8] | 2012 | Cross-sectional | 36 | Pearson | F | T | F | T | T | F | NA | NA |
| Winwood [45] | 2012 | Cross-sectional | 23 | Pearson | F | F | F | F | NA | F | NA | NA |
| Ye [47] | 2013 | Cross-sectional | 20 | Pearson | F | T | Partial | F | NA | F | NA | NA |
| Akagi [39] | 2014 | Cross-sectional | 18 | Pearson | F | T | T | F | NA | F | NA | NA |
| Kerksick [32] | 2014 | Cross-sectional | 66 | Pearson | F | Partial | F | T | T | F | NA | NA |
| Schumacher [43] | 2016 | Cross-sectional | 34 | Pearson | F | Partial | F | F | NA | F | NA | NA |
| Loturco [49] | 2017 | Cross-sectional | 36 | F | NA | NA | T | T | T | F | NA | NA |
| Ferland [7] | 2020 | Cross-sectional | 9 | Pearson | F | Partial | F | F | NA | F | NA | NA |
| Reya [5] | 2021 | Cross-sectional | 13 | Pearson, Spearman | T | Partial | F | F | NA | F | NA | NA |
| Ferrari [6] | 2022 | Cross-sectional | 74 | Pearson | F | F | F | F | NA | F | NA | NA |
| Massini [41] | 2022 | Cross-sectional | 30 | Pearson | F | T | Partial | T | T | F | NA | NA |
| Zaras [44] | 2023 | Cross-sectional | 21 | Pearson | F | Partial | F | F | NA | F | NA | NA |
| Nacleiro [31] | 2017 | Experimental | 242 | F | NA | NA | F | T | T | F | NA | NA |
| Garcia-Ramos [48] | 2018 | Experimental | 30 | Pearson | F | F | F | F | NA | F | NA | NA |
| Perez-Castilla [42] | 2020 | Experimental | 20 | F | NA | NA | Partial | T | T | F | NA | NA |
| Garcia-Ramos [15] | 2021 | Experimental | 12 | F | NA | NA | T | T | T | F | NA | NA |
| Lee [51] | 2023 | Experimental | 20 | F | NA | NA | NA | F | NA | T | F | F |
| Mann [40] | 2012 | Prospective cohort | 289 | Pearson | F | F | F | F | NA | F | NA | NA |
| Latella [37] | 2018 | Retrospective cohort | 1368 | F | NA | NA | NA | F | NA | T | Cohen's *d* | T |
| Solberg [38] | 2019 | Retrospective cohort | 118 | F | NA | NA | NA | F | NA | T | Mean difference | T |
| *Para athlete studies* | | | | | | | | | | | | |
| Loturco [52] | 2019 | Cross-sectional | 17 | F | NA | NA | T | T | T | F | NA | NA |

*(Continued)*

**Table 6.** (Continued)

| Author | Year | Study design | Sample size, n | Correlation analysis | Correlation uncertainty | Correlation p-value | Scatter plot | Regression analysis | R-squared | Between group analysis | Group difference effect size | Group difference uncertainty |
|---|---|---|---|---|---|---|---|---|---|---|---|---|
| Hamid [10] | 2019 | Cross-sectional | 52 | Pearson | F | T | F | F | NA | F | NA | NA |
| Iturricastillo [13] | 2019 | Cross-sectional | 9 | Pearson | F | T | T | T | T | F | NA | NA |
| Romarate [14] | 2021 | Cross-sectional | 10 | Pearson | T | T | T | F | NA | F | NA | NA |
| Teles [33] | 2021 | Cross-sectional | 19 | F | NA | NA | NA | F | NA | T | Cohen's d | F |
| Lopes-Silva [35] | 2022 | Retrospective cohort | 40 | F | NA | NA | NA | F | NA | T | F | F |
| Severin [36] | 2023 | Retrospective cohort | 328 | F | NA | NA | T | T | T | F | NA | NA |
| Lopes-Silva [34] | 2023 | Retrospective cohort | 1889 | F | NA | NA | NA | F | NA | T | F | F |

Note. F = False, NA = Not applicable, Partial = Reported only for some associations, T = True

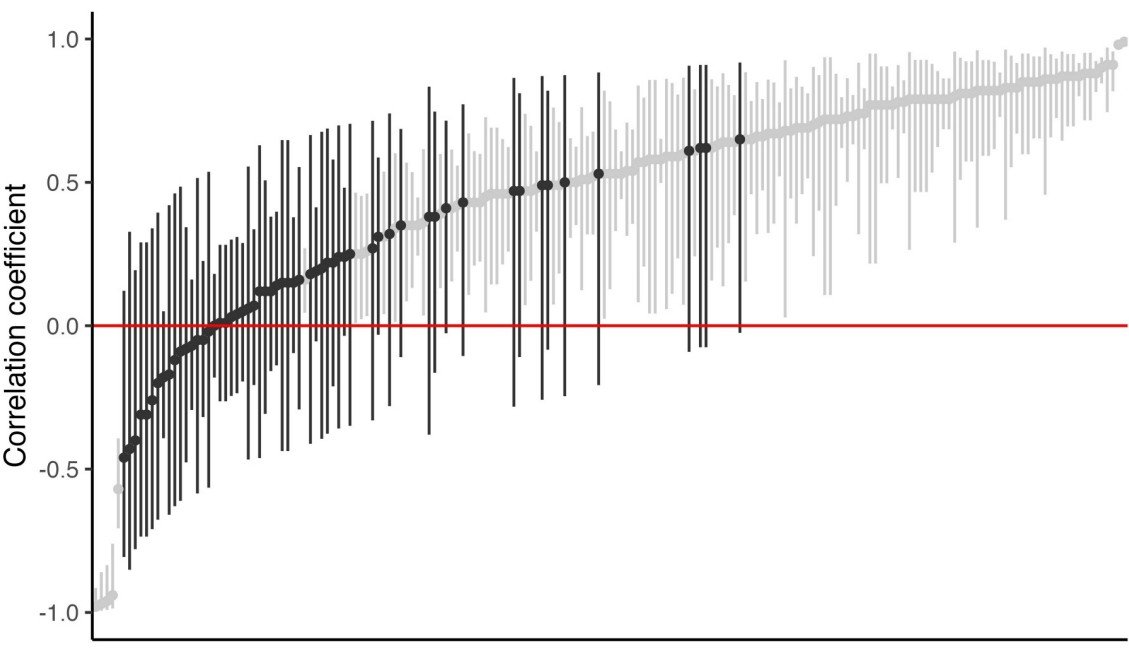

95% CI includes the null value of 0 ● True ● False

**Fig 4. Caterpillar plot of correlations (n = 183) between features of interest and one-repetition maximum bench press performance.** Closed circles indicate the point estimate, while error bars indicate the 95% confidence interval. Black intervals indicate instances when the 95% confidence interval included the null value of zero. The plot illustrates the influence of small study sample sizes on the uncertainty of correlation coefficients. Because study sample sizes were relatively small—median n of 30 for both non-disabled and Para athlete studies—there were instances where the point estimate was large enough to be of interest, but the 95% confidence interval included the null. For example, the confidence interval of 42.1% (16/38) of correlations that would commonly be interpreted as 'moderate' (i.e., r = −0.3 to −0.5 or r = 0.3 to 0.5) included the null.

## Discussion

There are seven key insights from our review of studies investigating features related to 1RM bench press performance. First, there was large heterogeneity in the specific variables studied. Second, few studies investigated features across multiple domains, for example, including anthropometric, body composition and technical features. Third, female participants were generally underrepresented. Fourth, information on participant impairment type and severity were often not reported by studies involving Para athletes. Fifth, there was an over reliance on bivariate correlation analysis, with relatively few studies considering a multivariable modelling approach. Sixth, despite using relatively small sample sizes, few studies considered the uncertainty of their results. Seventh, no studies considered open science or transparent research practices in the form of pre-registration, explicitly stating that they followed reporting guidelines, or sharing their data. Based on these findings, we make seven recommendations to guide future research, which are described in Table 7.

Studies involving Para athletes did not consistently report information on impairment type, origin of impairment, or if applicable, sport class. The organisation of Para powerlifting does not encourage differentiation between impairments in research—i.e., provided that Para athletes have one of the eight eligible impairments then they can compete in the appropriate sex and weight category. However, there is evidence that origin of impairment may influence bench press strength [34, 35], with higher strength values reported in athletes with acquired impairments compared to those with congenital impairments [33, 34]. It is possible that congenital impairments result in physiological changes that limit the potential for strength development, or that athletes with acquired impairments have high levels of strength development prior to their impairment, and these benefits persist post impairment. Irrespective, impairment type and origin of impairment are likely important variables that should be included in any Para sport transfer models, and therefore, need to be reported by studies. We recommend that future studies provide complete information on impairment type, origin of impairment, and sport class.

At the most recent Paralympic games in Tokyo, females accounted for 49.4% of 178 Para powerlifters [4]. Similarly, of the 196 entrants in the Tokyo Olympic weightlifting event, 49% were female [53]. Despite balanced participant rates at the elite level, we found that females were underrepresented across the 32 studies in the review (Table 1). The relative absence of

**Table 7. Recommendations for studies investigating features related to 1RM bench press performance.**

| Recommendation | Description |
|---|---|
| 1 | Consider training and non-training modifiable features across several domains of anthropometric, body composition, technical, and in Para athlete populations, disability impairment and type. |
| 2 | Include female athletes in studies, and report sample characteristics and study results separately for males and females. |
| 3 | When studying Para athletes, report information on impairment type and severity, origin of impairment, and where applicable, sport classification. |
| 4 | Combine features using a multivariable model approach, considering the potential influence of confounding, and modifying effects. |
| 5 | Preregister study aims, hypotheses and methods on a public platform, such as, on the Open Science Framework. |
| 6 | Use reporting checklists (https://www.equator-network.org/) to guide the communication of study methods and results. Emphasis should be placed on the uncertainty of results (e.g., a 95% confidence interval) when communicating study findings. |
| 7 | Share enough data to reproduce the key study findings, or a synthetic dataset, on a public platform |

female participants is disappointing but not surprising [54, 55]. We recommend that future studies include female participants, to better reflect the progress in female sport participation. It is reasonable to suggest that the specific features related to bench press strength may differ between the sexes [56]. The inclusion of females in future research will be essential to ensure that talent identification and sport transfer models are designed to cater for the unique and emerging cohorts of females in sport. We recommend that future works consider the effect of sex in any statistical modelling, and that study results be reported according to biological sex [57]. As per current consensus, when the effect of female sex hormones is not of interest, we recommend that studies report information on the menstrual status of participants, and a record of cycle or pill day taken, which can be collected via calendar tracking [57].

A total of 111 unique features from six domains (Fig 3) were investigated across the 32 studies in the review. Despite the wide range of unique features studied, only half of the studies in the review investigated features from different domains (Fig 3B). Features from multiple domains are likely important for bench press performance, and these are likely to be different for male and female athletes, and between non-disabled and Para athlete populations [56]. Talent identification and sport transfer models should consider features across multiple domains, and their interactions.

Features within the anthropometric and body composition domains demonstrated the strongest associations with bench press 1RM. In non-disabled athlete populations, body mass, arm circumference and cross-sectional area, triceps muscle thickness, and forearm, thigh, and calf circumference (Table 2), lean body mass, lean body mass to height ratio, body mass index and skeletal muscle mass (Table 3) were positively associated with bench press 1RM. In Para athlete populations, body mass, arm circumference and hip circumference were positively associated with bench press performance (Table 5), and there was evidence that origin of impairment was an important consideration [34, 35]. While features outside of the anthropometric and body composition domain are likely important, there was insufficient evidence to determine their association with bench press performance. The demographic feature of age was important in Para athlete populations (Table 5) but its association with bench press 1RM was unclear in non-disabled populations (Table 4). It is likely that age is a proxy of training history, and therefore, may not be a feature worth including in talent identification or sport transfer models.

Technical and neuromuscular features were least studied compared to other domains (Fig 3A). There was evidence that movement velocity at submaximal loads shared a strong association with bench press 1RM [15, 48, 58]. While load-velocity relationships may allow for the estimation of an individual's 1RM, the slope of the relationship appears unaffected by strength [59] and is unlikely to be useful in identifying talented lifters. Given that bench press and Para powerlifting are strength sports, the ability to produce high levels of voluntary force is important [5]. Neuromuscular variables provide insight into voluntary force production, and deficits in these variables may be associated with poorer performance in strength-based sports [60]. For instance, in Para powerlifting, athletes with neurological impairments, such as cerebral palsy or spinal cord injury, may have poor force producing capability compared to athletes with other types of impairments [61, 62]. Consequently, neuromuscular variables may be useful for developing talent transfer models, where the goal is to determine if Para athletes participating in other sports are good candidates for Para powerlifting. Many commonly studied features, such as arm circumference and arm cross-sectional area, are modifiable with training. From a talent identification perspective these features may offer limited information, as modifiable factors could be improved in a range of athletes and alone may not identify talented young athletes [63]. Modifiable factors may also be more useful in the context of Para powerlifting transfer, where promising athletes are identified within sports requiring similar qualities

[64]. We recommend that future works consider investigating a combination of modifiable and non-modifiable features, across several domains.

There was a primary reliance on bivariate correlation analysis (56% of studies) to investigate the association between features of interest and bench press 1RM. While correlation analysis is a useful exploratory data analysis tool, the development of talent identification and sport transfer models will require more advanced analytical methods. Multivariable modelling [65], confounding and effect modification [66], and non-linear modelling approaches, including tree-based methods, all require consideration. Future works should also consider using directed acyclic graphs to illustrate the assumptions about the causal structure of the relationships between the features being modelled [67]. We encourage researchers to engage a statistician or someone with statistical expertise [68].

We found that study sample sizes were generally small—the median sample size of both non-disabled and Para athlete studies was 30 individuals. Despite this, few studies considered the uncertainty of their results (Table 6). Fig 4 illustrates the potential problem with not considering the uncertainty of study results, with respect to correlation coefficients. There were instances where the point estimate was large enough to be of interest, but the 95% CI included the null value of zero (Fig 4). For example, there were 38 correlations that would commonly be interpreted as 'moderate', i.e., $r = -0.3$ to $-0.5$ or $r = 0.3$ to $0.5$ [69]; however, for 42.1% of these correlations, the confidence interval included the null value of zero, and therefore, no or only a 'trivial' association between the features of interest and bench press 1RM could not be ruled out. Failure to consider uncertainty of correlations may encourage performance staff to adopt weak evidence in their practice. For example, several trainable anthropometric factors have moderate correlation coefficients but their confidence intervals cross zero (see Fig 4). A focus on these factors may waste time that could be better spent focusing on other trainable factors underpinning performance. We recommend that future studies report results with a measure of uncertainty and considered the uncertainty when communicating the study results.

There was an absence of open science and transparent research practices across the 32 studies in the review. No studies were preregistered. Preregistration involves authors registering their aims, hypotheses, and methods on a publicly available platform, for example, the Open Science Framework [70]. Importantly, registration occurs before data collection. Preregistration helps guard against questionable research practices, including *p*-hacking, the post-hoc manipulation of hypotheses, and selectively discarding non-significant results [71]. No study shared their data, which is consistent with practices in the field [72]. Aside from data sharing improving the confidence in study results, it can help future studies overcome issues with small study sample sizes. Future works could use shared data in their analytical process, in the form of Bayesian informative prior distributions, which has the potential to improve the certainty of study results [73]. Finally, we found that no studies in the review indicated that they followed a reporting guideline. Following reporting guidelines improves research transparency. Guidelines for a range of study designs can be found on the Enhancing the Quality and Transparency of health Research (EQUATOR) network (https://www.equator-network.org/). We recommend that future studies preregister their aims, hypotheses, and methods, that authors share their data, and that reporting checklists are used to guide the reporting of study methods and results.

## Limitations

Several limitations of the review require acknowledgment. First, consistent with scoping review practices, we did not consider the quality of the 32 studies in the review. Second, we did not formally aggregate correlation coefficients using meta-analytical modelling. Our intention

was to summarise the features investigated, rather than comment specifically on the strength of associations between certain features and bench press 1RM. Nonetheless, several features that were investigated by multiple studies, such as chest circumference (Table 2), fat mass (Table 3) and age (Table 4), were deemed to have an unclear association with bench press 1RM. It is possible that formal aggregation of these studies by conducting a meta-analysis would show a positive association with bench press 1RM. Third, our review underestimates the number of females included in studies of non-disabled athletes, as female groups from three studies [7, 15, 31] were excluded after applying our inclusion criteria. Including the female groups from these three studies would have changed the proportion of females studied from 21.9% to 34.9%; however, it would not have changed our substantive conclusion that female athletes were generally underrepresented across the 32 studies in the review.

## Future directions

This review has highlighted several directions for future research. Firstly, longitudinal studies are needed to better understand which factors cause improvements in bench press performance. Prospective studies following athletes (non-disabled and Para) who progress from sub-elite to elite competition or remain in sub-elite competition would provide stronger evidence for the factors important for bench press performance than is currently available. Models of Para powerlifting performance based on an athlete monitoring system, where the bench press is performed regularly as part of a standardized test battery, would also be useful for understanding which factors are important for performance. While analysis of competition results provides some useful information, to progress our understanding of the development of bench press performance in Para athletes, empirical studies are needed. Longitudinal studies and models based on athlete monitoring systems would help identify important non-modifiable performance factors and benefit talent identification.

There is a need to better understand the neuromuscular factors associated with bench press performance. These variables were one of the most uncommonly studied and may be particularly relevant for Para athletes. Many impairments (e.g. spinal cord injury, cerebral palsy) impact motor unit recruitment and firing, which are important determinants of voluntary force production [61]. Other factors impacted by these impairments, such as patterns of inter-muscle coordination may also be important for performance in Para athletes, but our understanding of these factors is limited based on current work.

## Conclusion

Our review of studies investigating features related to bench press performance provides several insights. There was large heterogeneity in the specific variables studied, and studies to date have generally focused on features from one domain (e.g., anthropometric), rather than investigating features across multiple domains. We found preliminary evidence that anthropometric and body composition features were positively associated with bench press performance, in both non-disabled and Para athlete populations. Technical and neuromuscular features were relatively understudied, limiting any conclusions that can be made about their impact on performance at present. We highlight the need for future studies to investigate features from multiple domains, using multivariable modelling approaches that consider how features may interact. We also highlight the need for practices of open science and transparent research. Large longitudinal studies that use information from athlete monitoring databases are likely needed to better understand the specific features associated with bench press performance, and for the development of talent identification and sport transfer models.

## Supporting information

**S1 File. Systematic search strategy.** The systematic search strategy used to search scientific databases.
(DOCX)

**S1 Checklist. Preferred Reporting Items for Systematic reviews and Meta-Analyses extension for Scoping Reviews (PRISMA-ScR) checklist.**
(DOCX)

## Author Contributions

**Conceptualization:** Rob Buhmann, Mark Sayers, David Borg.

**Data curation:** Rob Buhmann, Julia O'Brien, David Borg.

**Formal analysis:** David Borg.

**Funding acquisition:** Rob Buhmann, Mark Sayers, David Borg.

**Investigation:** Rob Buhmann, Julia O'Brien, David Borg.

**Methodology:** Rob Buhmann, Mark Sayers, David Borg.

**Project administration:** Rob Buhmann.

**Resources:** Rob Buhmann, David Borg.

**Software:** Rob Buhmann, Julia O'Brien, David Borg.

**Supervision:** Mark Sayers.

**Visualization:** David Borg.

**Writing – original draft:** Rob Buhmann, Mark Sayers, David Borg.

**Writing – review & editing:** Rob Buhmann, Mark Sayers, Julia O'Brien, David Borg.

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
