## [Decision Letter · Decision Letter 0]

25 Jun 2024

PONE-D-24-20584Important Features of Bench Press Performance in Non-Disabled and Para Athletes: A Scoping ReviewPLOS ONE

Dear Dr. Buhmann,

Thank you for submitting your manuscript to PLOS ONE. After careful consideration, we feel that it has merit but does not fully meet PLOS ONE’s publication criteria as it currently stands. Therefore, we invite you to submit a revised version of the manuscript that addresses the points raised during the review process.

 All reviewers found merit in your manuscript but have offered suggestions for improvement. Please make sure you address the concerns related to clarity of methods and ensuring the discussion is linked to the original hypothesis.

We look forward to receiving your revised manuscript.

Kind regards,

Jeremy P Loenneke

Academic Editor

PLOS ONE

Journal Requirements:

   "Funding from the Australian Sports Commission"

Reviewers' comments:

Reviewer's Responses to Questions

**Comments to the Author**

1. Is the manuscript technically sound, and do the data support the conclusions?

Reviewer #1: Partly

Reviewer #2: Yes

Reviewer #3: Yes

2. Has the statistical analysis been performed appropriately and rigorously? 

Reviewer #1: N/A

Reviewer #2: Yes

Reviewer #3: Yes

3. Have the authors made all data underlying the findings in their manuscript fully available?

Reviewer #1: Yes

Reviewer #2: Yes

Reviewer #3: Yes

4. Is the manuscript presented in an intelligible fashion and written in standard English?

Reviewer #1: Yes

Reviewer #2: Yes

Reviewer #3: Yes

5. Review Comments to the Author

Reviewer #1: Manuscript Review: PONE-D-24-20584

Manuscript Summary:

The study analyzes factors influencing bench press performance in non-disabled athletes and Para athletes, aiming to provide insights for talent identification and transfer models in weightlifting. The hypothesis posits significant differences in bench press performance determinants between these two groups, influenced by variables such as age, sex, body mass, height, and muscular strength.

Main Findings and Highlights:

- Key determinants of bench press performance in non-disabled athletes include age, sex, body mass, height, and muscular strength.

- For Para athletes, adaptations due to disability, such as balance, stability, and modified technique, are crucial.

- The study discusses the implications for personalized talent identification and training programs in weightlifting for different athlete groups.

Points to Consider:

1) Grammatical or Agreement Errors:

Consistency in Verb Tenses: Review the entire manuscript to ensure consistency in the use of verb tenses.

Subject-Verb Agreement: Ensure proper agreement between subjects and verbs. For instance, on page 40, paragraph 1, line 1, the sentence could be reviewed for consistency.

2) Contrasting Content:

Clarity: Some sections might present contradictory or confusing information. A thorough review and clarification of these points are recommended.

3) Inconsistent Results:

Justification of Discrepancies: Some results appear inconsistent with the initial hypotheses or existing literature. These discrepancies should be clearly reviewed and justified within the manuscript.

4) Argumentative and Scientific Rhetoric Improvements:

Strengthen Arguments: Improve the connection between results and conclusions to ensure a solid and consistent argument.

Technical Language: Use more precise and technical language when describing methods and results to avoid ambiguities.

5) Reader Engagement:

Practical Examples: Incorporate practical examples or case studies to illustrate the applicability of the results in sports practice, making the manuscript more attractive to readers interested in bench press performance.

Hypothesis Confirmation:

Based on the main conclusions, the initial hypothesis does not appear objectively (I suggest it appears, it makes everything clearer!), however, implicitly the implied hypothesis seems partially confirmed. Significant differences were found in bench press performance determinants between non-disabled athletes and Para athletes, particularly regarding age, sex, body mass, height, and muscular strength. However, the study identified areas where results did not fully align with the hypothesis, such as the lack of comprehensive studies across multiple domains and the underrepresentation of female participants.

Conclusion Review:

The conclusion partially addresses the study's objectives. It provides valuable insights into bench press performance characteristics but could more explicitly link results to the study's initial objectives. Emphasizing how the findings contribute to understanding performance determinants and informing talent identification and transfer models in weightlifting would enhance the conclusion.

Recommendations:

Grammar and Agreement: Conduct a thorough grammatical review, focusing on verb tense consistency and subject-verb agreement.

Clarify Content: Address and resolve any contradictory information to ensure clarity.

Justify Inconsistencies: Clearly explain any discrepancies between results and initial hypotheses.

Strengthen Arguments: Enhance the connection between results and conclusions with precise, technical language.

Engage Readers: Use practical examples or case studies to illustrate the real-world applicability of the findings.

Conclusion Enhancement: Directly link the results to the study's objectives, providing a robust synthesis of findings.

By addressing these points, the manuscript will improve in clarity, cohesion, and impact, making it more informative and engaging for readers interested in bench press performance in non-disabled and Para athletes.

Reviewer #2: This scoping review explored variables related to bench press performance (1RM) in both able-bodied and para-athlete populations. Of the 32 studies included, the authors noted body composition and anthropometric variables had the strongest association with bench press performance. The study is well-written and fills an important gap in the literature. I have minor specific comments below.

1) Abstract– If character limits allow, could the authors identify all six of the domains into which features were grouped?

2) Introduction – the introduction is concise, references relevant literature, and provides the rationale for the current study.

3) Methods, exclusion criteria – It seems that excluding those who cannot bench press their body weight (especially for women) artificially limits the ranges of bench press performance for analysis. To me, this could limit valuable information on the predictors of performance if those with lower performance are not included. Could the authors include an analysis without this exclusion criterion (being able to bench press body mass) to see if the analysis changes?

4) Methods, selection criteria – was supportive equipment (e.g. bench shirt or wrist wraps) taken into account during 1RM assessment in the studies included? Please clarify.

5) Table 2 – waist circumference is listed twice (under upper limb and trunk categories) with slightly different correlation coefficients reported from the same study (reference #6). Please clarify.

6) Figure 4 is not brought up in the manuscript until the discussion. In my opinion, a description of this figure should first be mentioned in the results section, with commentary on the implications of this in the discussion.

7) The discussion proposes sound recommendations for future research based on the previous studies reviewed.

Reviewer #3: As per notes attached.

General comments:

Title

Are presented satisfactorily.

Abstract

It is written in a structured way, however, the methodology is written in a very summarized way which ends up making the findings and conclusions of the article.

It would be very important to have more details on how the study was carried out, such as query words, length of time it was consulted, that is, articles published within a certain period of time.

Furthermore, more numerical data should be presented in order to provide an initial overview of the manuscript.

Please confirm that the keywords are listed as descriptors in health sciences.

Introduction

Although the introduction is of a very good length, and initially with a contextualization of Powerlifting, whether conventional or palympic, and more specifically the bench press. However, the introduction cannot move from the general to the specific, that is, the association between performance and anthropometric characteristics associated with health. And on the other hand, the problem is not clear, not allowing us to understand the objectives, much less justify them. It would also be important to present research hypotheses to be answered.

Methods

It should present more clearly the design of the study.

The keywords researcher and the period of time researched are not included in the manuscript. As it turns out, the database and the language, therefore, the methodology is not complete, it would be good to complement it.

Results

Are presented satisfactorily.

Discussion

It should reaffirm the objectives and start discussing the results in the chronological order that appear in the item results.

Please present the limitations of the study.

Conclusion

Are presented satisfactorily.

References

Are presented satisfactorily, review their formatting. Of the 71 references, 37 are current and 34 have been published for more than five years. In this sense, it would be good to check current studies and, if necessary, include them in the manuscript.

Overview

The manuscript presented addresses a relevant research topic.

It would be advisable to do a general review.

6. PLOS authors have the option to publish the peer review history of their article (what does this mean?). If published, this will include your full peer review and any attached files.

Reviewer #1: No

Reviewer #2: No

Reviewer #3: **Yes: **Felipe J. Aidar

---

## [Author Response · Author response to Decision Letter 0]

12 Aug 2024

We have attached a word document containing the responses to reviewers with this submission.

---

## [Decision Letter · Decision Letter 1]

26 Aug 2024

Important Features of Bench Press Performance in Non-Disabled and Para Athletes: A Scoping Review

PONE-D-24-20584R1

Dear Dr. Buhmann,

We’re pleased to inform you that your manuscript has been judged scientifically suitable for publication and will be formally accepted for publication once it meets all outstanding technical requirements.

Kind regards,

Jeremy P Loenneke

Academic Editor

PLOS ONE

Additional Editor Comments (optional):

Reviewers' comments:

Reviewer's Responses to Questions

**Comments to the Author**

1. If the authors have adequately addressed your comments raised in a previous round of review and you feel that this manuscript is now acceptable for publication, you may indicate that here to bypass the “Comments to the Author” section, enter your conflict of interest statement in the “Confidential to Editor” section, and submit your "Accept" recommendation.

Reviewer #2: All comments have been addressed

2. Is the manuscript technically sound, and do the data support the conclusions?

Reviewer #2: Yes

3. Has the statistical analysis been performed appropriately and rigorously? 

Reviewer #2: N/A

4. Have the authors made all data underlying the findings in their manuscript fully available?

Reviewer #2: Yes

5. Is the manuscript presented in an intelligible fashion and written in standard English?

Reviewer #2: Yes

6. Review Comments to the Author

Reviewer #2: (No Response)

7. PLOS authors have the option to publish the peer review history of their article (what does this mean?). If published, this will include your full peer review and any attached files.

Reviewer #2: No

---

## [Editor Report · Acceptance letter]

5 Sep 2024

PONE-D-24-20584R1 

PLOS ONE

Dear Dr. Buhmann, 

I'm pleased to inform you that your manuscript has been deemed suitable for publication in PLOS ONE. Congratulations! Your manuscript is now being handed over to our production team.

Kind regards, 

on behalf of

Dr. Jeremy P Loenneke 

Academic Editor

PLOS ONE